# ARIES: A Novel Multivariate Intrusion Detection System for Smart Grid

**DOI:** 10.3390/s20185305

**Published:** 2020-09-16

**Authors:** Panagiotis Radoglou Grammatikis, Panagiotis Sarigiannidis, Georgios Efstathopoulos, Emmanouil Panaousis

**Affiliations:** 1Department of Electrical and Computer Engineering, University of Western Macedonia, 50100 Kozani, Greece; pradoglou@uowm.gr; 20INF, Imperial Offices, London E6 2JG, UK; george@0infinity.net; 3Department of Computing and Information Systems, University of Greenwich, Old Royal Naval College, London SE10 9LS, UK; e.panaousis@gre.ac.uk

**Keywords:** cybersecurity, Intrusion Detection System, Machine Learning, Modbus, SCADA, Smart Grid

## Abstract

The advent of the Smart Grid (SG) raises severe cybersecurity risks that can lead to devastating consequences. In this paper, we present a novel anomaly-based Intrusion Detection System (IDS), called ARIES (smArt gRid Intrusion dEtection System), which is capable of protecting efficiently SG communications. ARIES combines three detection layers that are devoted to recognising possible cyberattacks and anomalies against (a) network flows, (b) Modbus/Transmission Control Protocol (TCP) packets and (c) operational data. Each detection layer relies on a Machine Learning (ML) model trained using data originating from a power plant. In particular, the first layer (network flow-based detection) performs a supervised multiclass classification, recognising Denial of Service (DoS), brute force attacks, port scanning attacks and bots. The second layer (packet-based detection) detects possible anomalies related to the Modbus packets, while the third layer (operational data based detection) monitors and identifies anomalies upon operational data (i.e., time series electricity measurements). By emphasising on the third layer, the ARIES Generative Adversarial Network (ARIES GAN) with novel error minimisation functions was developed, considering mainly the reconstruction difference. Moreover, a novel reformed conditional input was suggested, consisting of random noise and the signal features at any given time instance. Based on the evaluation analysis, the proposed GAN network overcomes the efficacy of conventional ML methods in terms of Accuracy and the F1 score.

## 1. Introduction

The Smart Grid (SG) as the convergence of the electrical system engineering with the Information and Communication Technology (ICT) is expected to eliminate the significant limitations and shortcomings of the existing electrical grid, such as the energy conversation, the demand response and the optimal utilisation of the various assets. It is worth noting that the existing, conventional electrical grid transforms only the 1/3 of the fuel energy to electricity while 8% of the production is consumed along the transmission lines and 20% of the generation capacity is used to meet the peak demand [1]. The primary role of SG is to provide the utility companies with full visibility and pervasive control services as well as new communication capabilities in order to interact with each other and realise electricity transactions across the grid. Although this new reality will enable the power utilities to introduce an intelligent layer over their existing infrastructure and mechanisms, it raises severe cybersecurity issues and challenges that can generate domino effects and disastrous consequences in the overall electrical grid [2,3].

In focusing on the cybersecurity aspect, SG encloses both cybersecurity issues and vulnerabilities arising from the electrical engineering domain as well as the ICT field. In particular, electrical engineering processes rely on legacy industrial devices, such as Supervisory Control and Data Acquisition (SCADA) systems that do not include the sufficient authentication and authorisation mechanisms since the corresponding communication protocols have not been developed having cybersecurity in mind. Characteristic examples are Modbus, Distributed Network Protocol 3 (DNP3), IEC 60870-5-104, Generic Object Oriented Substation Events (GOOSE), Manufacturing Message Specification (MMS) and Profinet [4]. On the other side, concerning the ICT domain, its heterogeneous and independent nature creates many cybersecurity issues. First, SG includes many networks, such as Home Area Networks (HANs), Neighbour Area Networks (NANs) or Residential Area Networks (RANs) and Wide Area Networks (WANs) characterised by different attributes and hence different cybersecurity issues. Moreover, the presence of the Internet of Things (IoT) generates multiple concerns [5]. First, IoT relies on the Internet, which is an insecure environment by itself. Second, IoT includes a variety of communication means followed by the corresponding cyberthreats. Besides, the vast amount of IoT data constitutes an attractive target for the cybercriminals. Finally, the autonomous nature of the IoT devices to communicate with each other without human intervention increases significantly the cybersecurity concerns.

Both academia and industry have identified novel authentication and access control mechanisms that can enhance the overall security and safety of SG [6,7,8]. In particular, the IEC 62351 standard [9] defines appropriate solutions aiming to meet the security gaps of the vulnerable SCADA communication protocols. Although these solutions may be efficient, their implementation and validation in real conditions is a challenging step since the procedures of the electrical grid should operate continuously. Furthermore, the business requirements of several vendors and manufacturers hinder the adoption of these solutions. Moreover, it is worth mentioning that the Denial of Service (DoS) attacks remain a significant issue. Therefore, an effective countermeasure against the various cyberattacks is the timely detection. The main goal of an Intrusion Detection System (IDS) is to detect or even prevent possible cyberattacks timely without affecting the normal operation of the monitored system. In addition, a significant benefit of IDS is its ability to detect zero-day cyberattacks and unknown anomalies, utilising Machine Learning (ML) and Deep Learning (DL) methods.

In this paper, a novel anomaly-based IDS called ARIES (smArt gRid Intrusion dEtection System) is presented. The proposed IDS adopts a set of ML and DL methods and consists of three detection layers, namely (a) network flow-based detection, (b) packet-based detection, and (c) operational data-based detection. In particular, the first layer relies on network flow statistic features and detects DoS attacks, Secure Shell (SSH) brute force attacks, File Transfer Protocol (FTP) brute force attacks, port scanning attacks and bots. The second one inspects Modbus/Transmission Control Protocol (TCP) packets and analyses their attributes, thus recognising Modbus/TCP-related anomalies like unauthorised Modbus/TCP commands and function code enumeration attacks. Finally, the last layer detects potential anomalies upon operational data, such as generator motor voltage, generator motor speed and exciter motor voltage. It is noteworthy that real datasets originating from a power plant in Greece were used for the development of all ML/DL-based detection layers. Finally, concerning the operational data-based detection (third detection layer), a novel Generative Adversarial Network (GAN) [10,11] called ARIES GAN was implemented, which overcomes the detection performance of conventional ML methods. More detailed, concerning the construction of ARIES GAN, a balanced and normalised time series dataset with electricity measurements was used, applying the min-max scaler. The novel loss function was selected, taking into account the feature matching loss.

The rest of this paper is structured as follows. Section 2 describes relevant papers and lists the contributions of our work. Next, Section 3 presents the ARIES architecture and Section 4 analyses ARIES GAN. Finally, Section 5 is devoted to the ARIES evaluation and Section 6 concludes this work.

## 2. Related Work and Contributions

This section first analyses relevant IDS devoted to protecting SG, and secondly, it introduces the contributions of this work. In particular, the first subsection focuses on pertinent previous works, where each paragraph describes a different case. Next, the novelties of this paper are listed, taking into account the brief analysis of the relevant works in this field.

### 2.1. Related Work

Several works examine the security issues of SG [6,8,12,13,14,15,16,17,18]. In our previous work in [17], we provide a comprehensive survey concerning the IDS systems for SG. First, this study provides the necessary background of the SG architectural elements and their communications. Next, a thorough overview of IDS systems follows, analysing their architecture as well as the potential detection categories: (a) signature-based detection, (b) anomaly-based detection and (c) specification-based detection. Subsequently, the requirements of IDS for SG are presented, and a detailed analysis of relevant 37 IDS follows. In particular, the examined IDS focus on protecting (a) the entire SG ecosystem, (b) Advanced Metering Infrastructure (AMI), (c) substations, (d) synchrophasors and (e) SCADA systems. Based on this analysis, the shortcomings and limitations of the current IDS are provided, and possible research directions in this field are provided.

In [19], Igbe et al. present an anomaly-based IDS devoted to protecting the DNP3 communications. The proposed IDS is composed of three main blocks, namely (a) packet capture block, (b) preprocessing block and (c) deterministic Dendritic Cell Algorithm (dDCA) signal processing block. The first one is responsible for capturing the DNP3 network packets; the second one preprocesses the data coming from the first block, and the third block undertakes the detection process by implementing dDCA. To evaluate the performance of their IDS, the authors created an artificial dataset composed of Man-in-The-Middle (MiTM) attacks, DNP3 packet modification and injection attacks, DNP3 disable unsolicited messages attacks, DNP3 cold restart message attacks and Distributed Denial of Service Attacks (DDoS). Moreover, the authors indicate many features that can be used to detect anomalies concerning the DNP3 communications. Finally, Receiver Operating Characteristics (ROC) curves are used to assess the efficacy of the proposed IDS.

In [20], Manso et al. present a Software-Defined Networking (SDN) [21] based IDS capable of detecting and mitigating DDoS attacks. Their implementation belongs to the signature-based IDS [22] family and uses SDN to prevent/mitigate DDoS attacks. The detection process relies on the signature rules of Snort [23]. On the other side, the SDN architecture consists of the Mininet simulator [24] and the Ryu controller [25]. Mininet simulates a plethora of SDN switches and hosts, while Ryu controls the SDN switches. When an attack is detected, a corresponding signal is transmitted to Ryu, which then rearranges the network flows, utilising the respective OpenFlow [26] commands. The evaluation analysis relies on two attack scenarios that were emulated and detected successfully. Finally, the mitigation time is also calculated under different conditions.

In [27], Milajerdi et al. present HOLMES, a system capable of recognising Advanced Persistent Threats (APTs) [28]. In particular, HOMLES focuses on three aspects in detecting APT campaigns, namely (a) alert generation, (b) alert correlation and (c) attack scenario presentation. First, HOMLES analyses the activities taking place at a host level, aiming to generate alerts that are closely related to Tactics, Techniques and Procedures (TTP) of APT actors. Next, HOMLES investigates information flows among the various system entities, such as files and processes in order to correlate alerts. Finally, HOLMES provides a high-level scenario graph, which combines TTP and the information flows of the previous two steps. TTP are represented as nodes, while the information flows constitute the edges. Three techniques were adopted to minimise potential false positives, namely (a) avoiding spurious dependencies, (b) noise reduction and (c) signal correlation and detection. The evaluation of HOMLES was conducted, utilising data of the Defence Advanced Research Projects Agency (DARPA) computing program. Specifically, HOLMES was evaluated against nine real-life APT scenarios, showing its efficacy to discriminate benign and malicious activities.

Chekired et al. in [29] present a hierarchical and distributed IDS against false metering attacks. The proposed IDS adopts a fog computing [30] architecture, which is adapted appropriately in AMI. In particular, AMI is re-formed in three layers: (a) AMI layer, (b) fog layer and (c) cloud layer. For each layer, several IDS are deployed that cooperate with each other in a hierarchical manner. Therefore, based on the proposed fog computing architecture, there are (a) HAN IDS, (b) RAN IDS and (c) Fog IDS. The problem of intrusion detection is modelled as a stochastic Markov Chain, where the smart meters are classified into five states, namely (a) authentic (b) suspicious_min, (c) suspicious_max, (d) malicious_min and (e) malicious_max according to their measurements. For each state, specific threshold values are defined. The efficiency of the specific IDS is demonstrated, utilising extensive simulations with various performance metrics and real electricity data originating from the city of Toronto.

Xie et al. in [31] present Pagoda, a hybrid real-time provenance-based IDS. Provenance or differently the origin of data is represented by a Directed Acyclic Graph (DAG), which defines how the data was produced and how it comes to the current state. The nodes depict the examined objects, while the edges interpret the dependencies among them. As provenance is adopted in several domains (e.g., search, experimental document and security), various provenance processing systems have been developed, such as PASS [32], SPADE [33], Hi-Fi [34], and LPM [35]. Based on PASS and PIDAS [36] (a previous IDS developed by the same authors), Pagoda recognises intrusions by calculating the anomaly degree of not a single provenance path but also of the overall graph. In particular, its operation consists of three main steps. First, it calculates the anomaly degree of each path. Second, it multiplies the anomaly degree of each path with the path length. Finally, it computes the sum of these weights and divides it with the sum of the lengths of all paths. Thus, Pagoda detects whether an intrusion affects only one object (e.g., file) or a set of objects. The performance of Pagoda is confirmed under many applications, including samba, distcc and linux-apr13.

In [37], Albanese et al. present an evaluation framework that assesses whether the probability of a sequence of security events is unexplained. The security events can originate either from an intrusion detection or alert correlation processes. The proposed framework does not aim to replace the processes mentioned earlier, but operates on top of them, taking into account their outcomes and identifying what is not adequately explained. The evaluation analysis demonstrates the efficiency of the proposed framework in terms of accuracy.

In [38], Zhang et al. introduce a Backward Influence Factor (BIF) algorithm, which addresses potential issues coming from incomplete IDS alerts, such as a) high False Positives (FP), high redundancy and incomplete data. It is a sequence pattern mining algorithm, which analyses IDS logs in real-time based on five phases. The first phase called normalising is responsible for aggregating and normalising all the IDS alerts into a standard format. Next, the second phase groups the alerts based on two fields: the source and destination IP addresses. Thus, the repeated alerts are merged. Then, the intrusion actions are extracted based on two fields: (a) type and (b) destination port. The third phase is the intrusion session pruning phase, where an extensive intrusion action sequence can be separated into several sequences (i.e., intrusion sessions) based on the average time of each intrusion action. Next, the pruning deletes the repeated subpatterns from the original sequence. Next, the correlation discovery phase follows, where all pruned sessions are inserted into the correlation discovery module, which performs the BIF algorithm, thereby identifying the association rate among the intrusion sessions. Finally, the Dynamic Correlation Graph (DCG) is formed based on the outcome of the previous phase. According to the evaluation analysis, the proposed method efficiently discovers intrusion patterns.

Finally, in [39], Chakraborty, et al. present a deception system called FORGE, which is capable of protecting possible data breaches. For each real document d, FORGE generates a set of fake documents d that are very similar to the original d. Therefore, a potential attacker can violate a fake document instead of the real one, which can be translated into financial and time costs. Each document is represented as a Multi-Layer Graph (MLG). Then, the authors adopt a novel Meta-Centrality (MC) approach to calculate the significance of the document’s concepts in the MLG representation. The rationale is that an ontology can be utilised to replace some document’s concepts with other fictitious but plausible. In addition, the authors demonstrate that the generation of fake documents from a real one can be solved as an integer linear programming problem. Based on the experimental results, it is shown that FORGE can produce effectively convincing fake documents.

Undoubtedly, the aforementioned works offer valuable information and useful methodologies. In particular, each detection category includes the respective advantages and disadvantages. Signature-based IDS recognise known cyberattacks with high detection performance, but they cannot detect zero-day attacks. On the other side, anomaly-based IDS can detect zero-day attacks, but they usually present high False Positive Rate (FPR). Finally, although the specification-based IDS can detect zero-day attacks and present low FPR, in a dynamic environment like SG, the specification rules should be updated continuously. However, it should be noted that the detection mechanisms of many IDS described earlier rely on the network traffic characteristics of the network and transport layers, without taking into account possible cyberattacks taking place at the application layer protocols (e.g., Modbus, DNP3). Moreover, it is worth noting that most of the anomaly-based IDS utilise outdated publicly available datasets, such as KDD CUP 1999 and NSL-KDD [40,41]. These datasets were not created, considering the unique attributes of an SG environment; therefore, the detection mechanisms based on them cannot be considered as reliable.

### 2.2. Contributions

Taking into account the remarks mentioned above, Figure 1 presents the contributions of this paper. In particular, compared to the previous works, this paper provides three independent detection layers that can recognise more potential intrusions rather than the existing works and the conventional IDS that investigate only a single data aspect, for example, the detection based on network flows. Moreover, instead of similar ML-based works that use either outdated datasets or simulated environments, it is noteworthy, that the proposed IDS was developed and evaluated, using real data originating from an industrial environment. In particular, each kind of network presents the corresponding peculiarities. Different types and quantities of information are produced in Home Area Networks (HAN), Industrial Area Networks (IAN) and Business Area Networks (BAN). Consequently, it is critical to adapt the IDS systems with the corresponding characteristics and data; otherwise, they will not provide reliable outcomes under real conditions. Furthermore, to the best of our knowledge, we are the first that provide an IDS, which introduces a deep learning approach and particularly a GAN network, which processes electricity measurements coming from a real power plant in Greece to identify unknown anomalies either due to a security violation or an energy-related anomaly (e.g., electricity disturbances). To this end, the ARIES GAN was created capable of managing appropriately this data. According to the evaluation results, the performance of ARIES GAN is better than other ML and DL techniques. To conclude, the contributions of our work are summarised as follows:**Providing ARIES, a three detection layer IDS system for SG**: The proposed IDS consists of three detection layers: (a) network flow-based detection, (b) packet-based detection, and (c) operational data-based detection. The first layer focuses on network flow statistic features and can detect (a) DoS attacks, (b) SSH brute force attacks, (c) FTP brute force attacks, (d) port scanning attacks and (e) bots. The second layer is devoted to the Modbus/TCP packets by analysing their attributes. Thus, Modbus/TCP-related attacks, such as unauthorised Modbus commands and function code enumeration can be recognised. Finally, the last layer detects anomalies upon operational data that can originate either via an electricity-related disturbance or a cybersecurity policy violation.**ARIES GAN**: The main novelty of this paper is the development of a GAN network with novel error minimization functions, considering the reconstruction difference. Furthermore, a novel reformed conditional input was suggested, consisting of random noise and the signal features at any given time instance. Based on the evaluation results, the proposed GAN network overcomes the efficacy of other ML methods.**Development and Evaluation based on Real Data**: In contrast to similar works that use ML techniques, the proposed IDS was developed and evaluated utilising real normal data coming from a power plant in Greece. Thus, the ARIES applicability and validity was evaluated under real operational conditions, ensuring the detection performance.**Evaluating a plethora of ML/DL methods**: Multiple ML/DL methods were evaluated for each detection layer in terms of Accuracy, True Positive Rate (TPR), FPR and the F1 score. Each detection layer adopts the ML/DL model originating from the method with the highest F1 score.

## 3. ARIES Architecture

This section analyses the ARIES architecture. As depicted in Figure 2, three modules compose the ARIEC architecture, namely (a) Data Collection Module, (b) ARIEC Analysis Engine and (c) Response Module. The Data Collection Module is responsible for feeding the ARIES Analysis Engine with the necessary data. The ARIES Analysis Engine integrates the three detection layers of ARIES. Finally, the Response Module generates security events based on the outcome of the ARIES Analysis Engine. It should be noted that ARIES is a Network IDS (NIDS) capable of receiving the network traffic data generated in a subnet (i.e., ARIES is installed in a centralised server, which aggregates the network traffic generated by all devices utilising a Switched Port Analyser (SPAN) port). On the other side, Figure 3 summarises the technologies utilised by the aforementioned ARIES components. The following subsections detail each component.

### 3.1. Data Collection Module

The Data Collection Module undertakes to feed periodically the ARIES Analysis Engine with the necessary data for detecting intrusions. Three kinds of data are collected, namely (a) network flow statistical features, (b) Modbus/TCP network packets information and (c) operational data. Concerning the first category, the CICFlowMeter generator [42,43] is used. CICFlowMeter captures network traffic data and extracts bidirectional network flows, including 84 network flow-related statistic features, such as the number of packets, the number of bytes and the network flow duration. On the other side, the Scapy [44] library was used for sniffing and interpreting Modbus packets. Scapy constitutes a network packet manipulation tool, which supports multiple network protocols, and has the ability to sniff, forge and dissect network packets. Next, concerning the operational data, it is received through a web-based interface between the Data Collection Module and a centralised server, where this data is stored. It is noteworthy that all data is coming from a real power plant in Greece. Therefore, regarding the operational data, the following electricity measurements are collected 24 V Battery voltage, 60 V Battery voltage, generator, motor speed, generator motor voltage, generator motor current, exciter motor voltage, exciter motor current, and temperature of incoming cooling water. Finally, before feeding the aforementioned data to the ARIES Analysis Engine, the Data Collection Module undertakes to preprocesses it, applying the min-max scaler described by Equation (Equation 1).
(1)$$z=\frac{x-min(x)}{max(x)-min(x)}$$

### 3.2. ARIES Analysis Engine

As illustrated in Figure 2, the ARIES Analysis Engine is composed of three detection layers: (a) network flow-based detection layer, (b) packet-based detection layer and (c) operational data-based detection layer. The first detection layer receives the statistic features originating from CICFlowMeter [42,43] in the Data Collection Module and recognises possible cyberattacks. In particular, it is composed of two detection models: (a) Intrusion Detection Model (IDM) and (b) Anomaly Detection Model (ADM) that operate complementarily. In particular, first, IDM is activated, trying to identify a cyberattack related to a specific network flow. IDM can detect five cyberattacks: (a) DoS, (b) SSH brute-force attack, (c) FTP brute-force attack, (d) port scanning and (e) bot. Industrial devices, such as Programmable Logic Controllers (PLCs) and Remote Terminal Units (RTUs) are prone against the aforementioned cyberattacks. Specifically, a port scanning attack tries to gather information related to the target system. Usually, such cyberattacks compose the first step to execute the following ones. DoS and Bots target to terminate the availability of these devices. Finally, SSH and FTP brute-force attacks aim to violate the confidentiality and integrity of the target. If IDM recognises the network flow as normal, then ADM undertakes to identify whether an anomaly behaviour is related to this network flow or not. IDM relies on ML supervised detection methods and particularly a decision tree classifier [45]. On the other side, ADM is based on an autoencoder [46,47]. The classification efficiency of IDM and ADM is analysed in detail in Section 5.

The second detection layer receives the Modbus packet information coming from the Data Collection Module and identifies whether a Modbus/TCP packet is anomalous or not. The operation of the second detection layer relies on unsupervised detection and particularly on an ML model trained with the Isolation Forest method [48]. The efficiency of the second detection layer is also analysed in Section 5.

The third detection layer detects anomalies upon operational data received by the Data Collection Module. The operation of this detection layer relies on ARIES GAN. ARIES GAN is analysed in detail in Section 4 while its performance is detailed in Section 5.

### 3.3. Response Module

The Response Module receives the output of the ARIES Analysis Engine and generates security events based on the AlienVault Open Source SIEM (OSSIM) [49,50] format. Moreover, based on the information originating from the three detection layers of ARIES, it activates appropriate firewall rules to mitigate timely the potential intrusions. The iptables [51] firewall was adopted for this scope. In particular, based on the respective security events, the appropriate iptables rules are constructed, thus corrupting the network traffic generated between two endpoints specified by the IP addresses and the TCP/User Datagram Protocol (UDP) ports.

## 4. ARIES GAN

The aim of an adversarial network concerning the problem of the anomaly detection is to train an unsupervised network, which will be capable of recognising anomalies, using a dataset, which includes data of a single class. In particular, this data is used only for the training process and denotes a benign behaviour. Supposing that we possess two datasets: (a) a training dataset $D=\{X_{1},\ldots ,X_{M}\}$, which contains *M* normal occurrences and (b) a testing dataset $\hat{D}=\left\{\left({\hat{X}}_{1},y_{1}\right),\ldots ,\left({\hat{X}}_{N},y_{N}\right)\right\}$ which includes *N* both normal and abnormal occurrences and $y_{i}\in \left[0,1\right]$ denotes the label of each occurrence. It is worth noting that M≫N.

The goal is to model appropriately *D* to understand the manifold representation and then to recognise anomalies in $\hat{D}$. In particular, the model *f* learns the normal data distribution and produces an anomaly score A(x). A high value of A(x) denotes a potential anomaly for the specific data point. More precisely, a threshold value *T* determines whether A(x) indicates an anomaly or not if A(x)>t. *T* is defined experimentally, utilising a testing dataset.

Figure 4 depicts ARIES GAN architecture, which is composed of two sub-networks: (a) generator and (b) discriminator. The generator receives the input z={x(t),R} representation that includes the real data x(t) at the current time *t* and a noise vector *R*. The output $x^{\prime }$ is the reconstruction of the input data for the current time *t* and all the previous *N* instances. In particular, the generator *G* first reads the input *z*, where $z\in {ℜ}^{w\times 2}$, and forwards it to the encoder network *E*. Based on fully connected layers followed by a batch-norm and leaky ReLU() activation function, *G* regresses *z* to $x^{\prime }$. Consequently, the generator *G* produces the data $x^{\prime }$ via $x^{\prime }=G\left(z\right)$, where z={x(t),R}. On the other side, the discriminator intends to distinguish the input $\bar{x}$ and the output $x^{\prime }$ as real or fake, respectively. It consists of fully connected layers followed by batch-norm and leaky ReLU activation.

Supposing that abnormal data points are forward-passed into the network *G*; however, since the generator is modelled only with normal samples during the training, it fails to reconstruct the abnormalities in the previous *N* time instances. Here, we consider that abnormal data does not occur in single time instances. An output $x^{\prime }$ that has not taken into account anomalies can result in the encoder network *E* to correspond $x^{\prime }$ to a vector $z^{\prime }$ which also has not considered an anomalous feature representation, thereby creating a dissimilarity between *z* and $z^{\prime }$. In such a case, (i.e., when there is a dissimilarity within the latent vector space for an input signal x(t)), the model categorizes *x* as an anomaly.

Regarding the training process of this network, the loss function was selected, considering the feature matching loss as illustrated in Equation (Equation 2).
(2)$$L_{adv}={∥f\left(\bar{x}\right)-f\left(x^{\prime }\right)∥}_{2}$$In particular, *f* is a function, which outcomes an intermediate layer of the discriminator *D* based on a given input $\bar{x}$; feature matching calculates the $L_{2}$ distance between the feature representation of the real and the produced data points.

## 5. Evaluation Analysis

This section is devoted to the evaluation analysis, describing the environment used for the evaluation process, the datasets, the comparative ML/DL methods and the experimental results.

### 5.1. Evaluation Environment

The data used for the development and the evaluation of ARIES originate from a real power plant in Greece, which consists of multiple PLCs that monitor and control the functionality of the industrial equipment (e.g., transformers, valves and generators). Moreover, the power plant uses a set of smart meters to capture operational data. These measurements are stored in a centralised server, which also plays the role of a Master Terminal Unit (MTU), transmitting appropriate commands to PLCs via Modbus/TCP. By using SPAN, the entire network traffic is forwarded in a destination server where ARIES is installed. Thus, the Data Collection Module of ARIES can receive network flows and Modbus/TCP network packets. On the other side, the operational data collected in MTU is transmitted to ARIES via a web-based interface.

### 5.2. Datasets

The first detection layer of ARIES and particularly IDM relies on ML/DL supervised detection methods, which need a labelled dataset, including both normal and malicious flows. To this end, the CSE-CIC-IDS2018 [52] was used, including labelled network flow statistics related to the cyberattacks that can be detected by IDM. These statistics were combined with normal ones originating from the network traffic (e.g., pcap files) of the power plant. The entire dataset was balanced appropriately so that the training and the testing dataset include the same number of records for each class. It is noteworthy that the malicious network flow statistics extracted from CSE-CIC-IDS2018 [52] would be similar if not the same, whether we executed these cyberattacks against the power plant. On the other side, ADM is based on unsupervised/semi-supervised detection methods; therefore, a dataset composed only with normal network flow statistics is needed. Thus, the training dataset of ADM includes only normal network flow statistics coming from the power plant. However, regarding the testing dataset of ADM, it consists of both normal and abnormal records. The anomalous records originated again from CSE-CIC-IDS2018 [52], by changing the corresponding cyberattack label to “Anomaly”. It should be noted that also, in this case, the testing dataset was balanced, including the same number of normal and abnormal records. In all cases, CICFlowMeter [42,43] was used for extracting the network flow statistics. In particular, from the 84 statistic features generated by CICFlowMeter, the following ones were used after a Principal Component Analysis (PCA) [53] processing.

Flow Duration: Identifies the duration of a network flow in microseconds.TotLen Fwd Pkts: Identifies the total size of packets to the forward direction.Fwd Pkt Len Mean: Denotes the average size of the packets to the forward direction.Fwd Pkt Len Mean: Denotes the average size of the packets to the forward direction.Bwd Pkt Len Std: Indicates the standard deviation size of the packets to the backward direction.Flow IAT Std: Implies the standard deviation time between two packets transmitted to the forward direction.Bwd Pkts/s: Indicates the number of the packets sent to the backward direction per second.Subflow Bwd Pkts: Denotes the average number of packets in a subflow to the backward direction.Init Bwd Win Bytes: Identifies the number of bytes transmitted in an initial window to the backward direction.Active Mean: Indicates the average time of a network flow, which remained active before becoming idle.

The second detection layer of ARIES focuses on Modbus/TCP packets attributes and utilises unsupervised/semi-supervised detection methods. Therefore, the training dataset includes only normal records, while the testing dataset contains both normal and abnormal records. Also, in this case, the testing dataset was balanced, thereby containing the same number of normal and abnormal records. The following Modbus/TCP attributes compose the records of these datasets. They are extracted from Modbus/TCP network traffic data (i.e., pcap files) using a custom Scapy script [44].

TCP Len: Indicates the size of the TCP frame.Transaction ID: The Transaction ID is used for transaction pairing when multiple Modbus messages are sent during the same TCP connection by a client without waiting for a prior response.Protocol ID: The Protocol ID is always equal to zero for Modbus and other values are reserved for possible extensions.Len: Indicates the length of the Unit ID, Fcode and the Modbus payload data.Unit ID: Unit ID is used for serial bridging to identify a remote server on a non-Transmission Control Protocol/Internet Protocol (TCP/IP) network.Fcode: Indicates the Modbus function code. The function codes of Modbus correspond to specific operations that are defined in detail in [54].Starting Address: Denotes the starting address of coils or registers where the values will be requested.Byte Count: Specifies the size of the response Modbus packets.

Finally, the third detection layer adopts ARIES GAN, which belongs to the unsupervised detection methods. Therefore, the training dataset includes only normal instances, while the testing dataset contains an equal number of both normal and anomalous records. Each record consists of multiple raw energy measurements, such as generator motor voltage, generator motor speed and exciter motor voltage. These measurements are explicitly related to the examined evaluation environment (i.e., the specific power plant). The anomalous records were identified by specific experts. A sample of this dataset is illustrated in Figure 5.

### 5.3. Comparative ML/DL Methods

Several ML/DL methods were prepared and compared with each other for each ARIES detection layer based on the aforementioned evaluation metrics. In particular, regarding the first detection layer of ARIES (IDM), nine ML methods were utilised, namely (a) Logistic Regression [55], (b) Linear Discriminant Analysis (LDA) [56], (c) Decision Tree Classifier [57], (d) Naive Bayes [58], (e) Support Vector Machine (SVM) [59], (f) Random Forest [60], (g) Multi-Layer Perceptron (MLP) [61], (h) Adaboost [62] and (i) Quadratic Discriminant Analysis [63]. Moreover, two DL methods were used, including two custom dense deep classifiers, namely (a) Dense Deep Neural Network (DNN) Relu and (b) Dense DNN Tanh. Both of the custom DNNs enclose four hidden layers and use the Relu and Tanh activation functions, respectively. In the output layer, both of them use the softmax activation function. Furthermore, the sparse categorical cross-entropy and the Adadelta are employed for the loss and optimisation. Dense DNN Relu possesses approximately 1058 parameters, while Dense DNN Tanh includes 340 parameters.

On the other side, regarding ADM, the second and the third detection layer of ARIES, six outlier detection methods were applied and evaluated, including (a) Angle-Based Outlier Detection (ABOD) [64], (b) Isolation Forest [48], (c) PCA [65], (d) Minimum Covariance Determinant (MCD) [66], (e) Local Outlier Factor (LOF) [67] and (f) the DIDEROT autoencoder [47]. Moreover, the ARIES GAN was also included and evaluated in the comparison related to the third detection layer of ARIES since it is devoted to recognising anomalies exclusively upon operational data.

### 5.4. Evaluation Results

Before describing the metrics that will be used for assessing the efficiency of the ARIES detection layers, first, some essential terms should be explained. In particular, True Positives (TP) indicates the number of classifications that detected cyberattacks and/or anomalies successfully. True Negatives (TN) denotes the number of classifications that classified correctly the normal operations. On the other side, FP defines the number of wrong classifications that categorised the normal operations as malicious activities. Finally, False Negatives (FN) denotes the number of incorrect classifications that classified cyberattacks or anomalies as normal activities. Based on these terms, the following metrics are defined. Moreover, it is noteworthy that regarding the first detection layer of ARIES, the following metrics were calculated by measuring entirely TP, TN, FP and FN for each class.

Accuracy (Equation (Equation 3)) denotes the proportion between the correct predictions and the total number of samples. Accuracy can be used as an impartial measure when the training dataset consists of an equal number of data samples for all classes. For instance, if the training dataset includes 90% data samples that present normal behaviour and 10% data samples with anomalies, then the Accuracy can reach 90% by predicting each instance as normal.
(3)$$Accuracy=\frac{TP+TN}{TP+TN+FP+FN}$$FPR (Equation (Equation 4)) indicates the ratio of normal behaviours that were detected as intrusions. It is calculated by dividing FP with the sum of FP and TN.
(4)$$FPR=\frac{FP}{FP+TN}$$TPR (Equation (Equation 5) ) calculates what ratio of intrusions that truly present a malicious behaviour was classified as an intrusion. It focuses mainly on FN and is computed by dividing TP with the sum of FN and TP.
(5)$$TPR=\frac{TP}{TP+FN}$$Finally, the F1 score (Equation (Equation 6)) represents the golden ratio between the Precision and TPR, considering both FP and FN. Precision is another measure, which calculates the ratio of those data samples that are and were categorised as anomalous.
(6)$$F1=\frac{2\times Precision\times TPR}{Precision+TPR}\;\mathrm{where}\;Precision=\frac{TP}{TP+FP}$$

Table 1 summarises the evaluation results of the ARIES first detection layer and particularly IDM in terms of Accuracy, TPR, FPR and the F1 score. The Decision Tree Classifier presents the best results where Accuracy=0.994, TPR=0.982, FPR=0.003 and F1=0.982. Decision trees are effective ML mechanisms that can be used for classification and regression problems. They consist of internal nodes and leaves. Based on the training features, the internal nodes possess edges that divide the overall space into smaller sub-spaces related to the corresponding classes. Accordingly, leaves represent the classes of the classification problem. Therefore, a directed tree is formed where the classification process is conducted by following the decision tree paths. More precisely, each path corresponds to specific logical rules. In this paper, the proposed decision tree was implemented by using the Classification and Regression Tree (CART) method [57] and the Information Gain (IG) criterion. Figure 6 depicts the confusion matrix of the Decision Tree Classifier. On the other side, the Quadratic Discriminant Analysis presents the worst results where Accuracy=0.722, TPR=0.166, FPR=0.166 and F1=0.166. Regarding the deep learning methods, Dense DNN Relu presents Accuracy=0.984, TPR=0.954, FPR=0.009 and F1=0.954, while the performance of Dense DNN Tanh is reflected by Accuracy=0.965, TPR=0.897, FPR=0.020 and F1=0.897. It is noteworthy that all ML/DL methods were evaluated under the same circumstances and also they were fine-tuned, by experimentally selecting the appropriate values for the respective parameters.

Similarly, Table 2 summarises the evaluation analysis of ADM. Based on the comparative analysis, the best performance is carried out by the DIDEROT autoencoder where Accuracy=0.9514, TPR=1, FPR=0.0963 and F1=0.953. The analysis of the specific DNN is given in our previous work in [47]. Figure 7 illustrates the confusion matrix of the proposed autoencoder. The evaluation results of ABOD [64], LOF [67] and Isolation Forest [48] are also very close to those of the DIDEROT autoencoder. On the other hand, MCD [66] presents the worst results where Accuracy=0.493, TPR=0.001, FPR=0.097 and F1=0.002.

Table 3 presents the performance analysis of the second detection layer of ARIES. Isolation Forest [48] gives the best results in terms of all evaluation metrics. The Isolation Forest method [48] discovers anomalies by deliberately “overfitting” models that memorise each data point. As outliers have more empty space around them, they take fewer steps to memorise. The method uses full decision trees where each leaf is considered as a single data point and the path length between the leaves and the root is computed. For each data point, the final measure is the average path length. The anomalous data points should be distinguished so that the average path should be relatively short. On the other side, the worst F1 score, FPR and TPR are presented by MCD [66] and LOF [67] where both of them reach 0.240, 0.135 and 0.210 respectively. Moreover, regarding DL, in this case the DIDEROT autoencoder reaches Accuracy=0.746, TPR=0.978, FPR=0.311 and F1=0.607. It is worth mentioning that as in the previous cases, all methods were evaluated under the same circumstances.

Table 4 portrays the evaluation of the third ARIES detection layer. ARIES GAN presents the best outcome where Accuracy and the F1 Score reach to 0.930 and 0.853, respectively. Figure 8 illustrates the confusion matrix of the ARIES GAN.

## 6. Conclusions

The rapid and continuous progression of SG requires appropriate solutions that will be able to address efficiently the various cybersecurity-related challenges. Possible invasions against SG can cause devastating consequences; therefore, the presence of appropriate intrusion detection mechanisms capable of detecting potential cyberattacks and anomalies against both industry and ICT systems is necessary.

This paper presents ARIES, an anomaly-based IDS system capable of protecting SG, by combining three detection layers: (a) network flow-based detection, (b) packet-based detection and (c) operational data-based detection. For each of the previous layers, multiple ML/DL methods were evaluated, utilising real data originating from a power plant in Greece. Regarding the third detection layer, a novel GAN network named ARIES GAN was proposed, which overcomes the performance of the conventional ML methods.

Our future plans related to this paper include the development of correlation mechanisms that will combine the results of the three ARIES detection layers, thus eliminating possible FP and identifying new security events. Moreover, more industrial and IoT protocols, such as IEC 60870-5-104, IEC 61850 and DNP3 will be investigated, thus detecting relevant attacks.

## Figures and Tables

**Figure 1 sensors-20-05305-f001:**
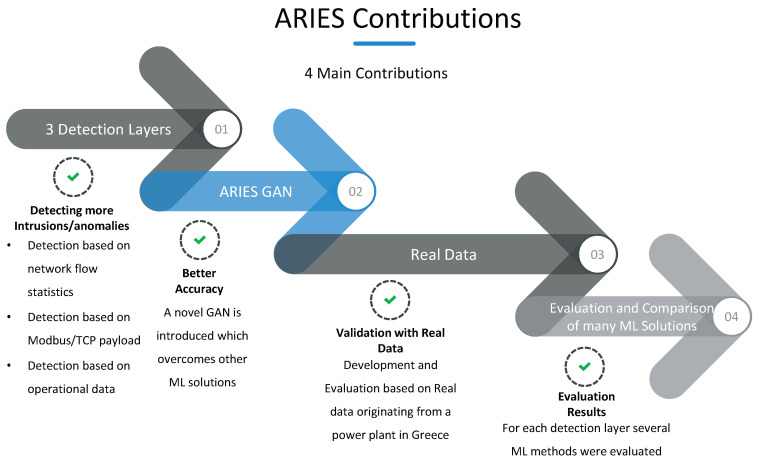
ARIES Contributions.

**Figure 2 sensors-20-05305-f002:**
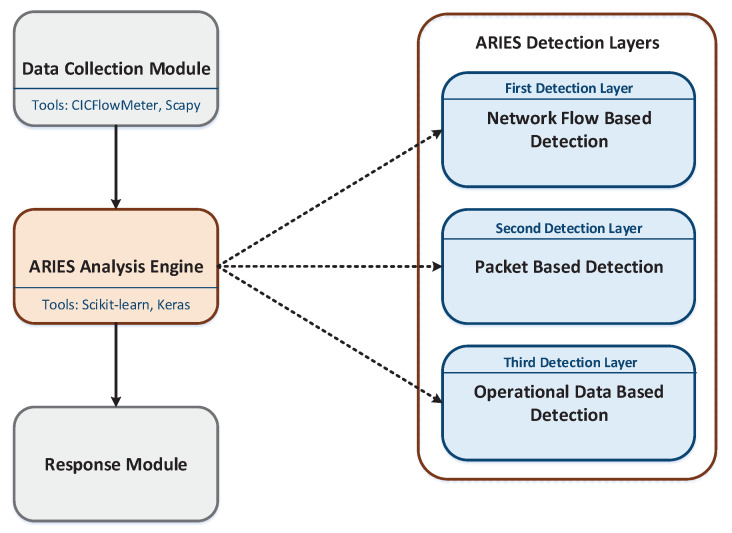
ARIES Architecture.

**Figure 3 sensors-20-05305-f003:**
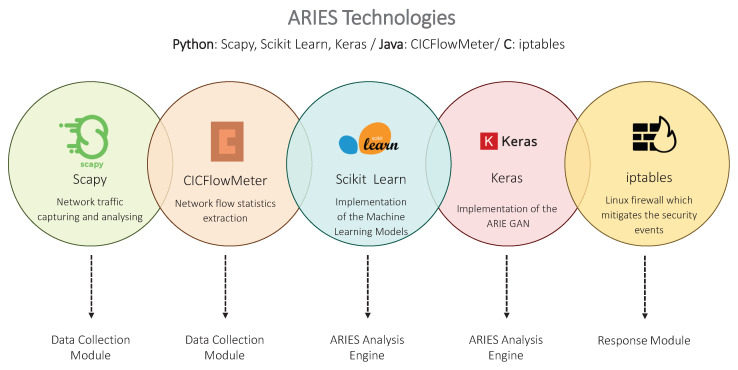
ARIES Technologies.

**Figure 4 sensors-20-05305-f004:**
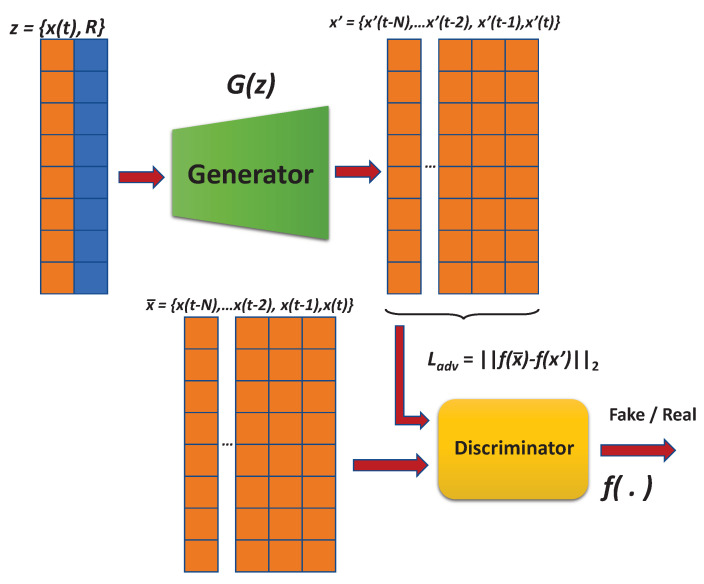
ARIES GAN architecture.

**Figure 5 sensors-20-05305-f005:**
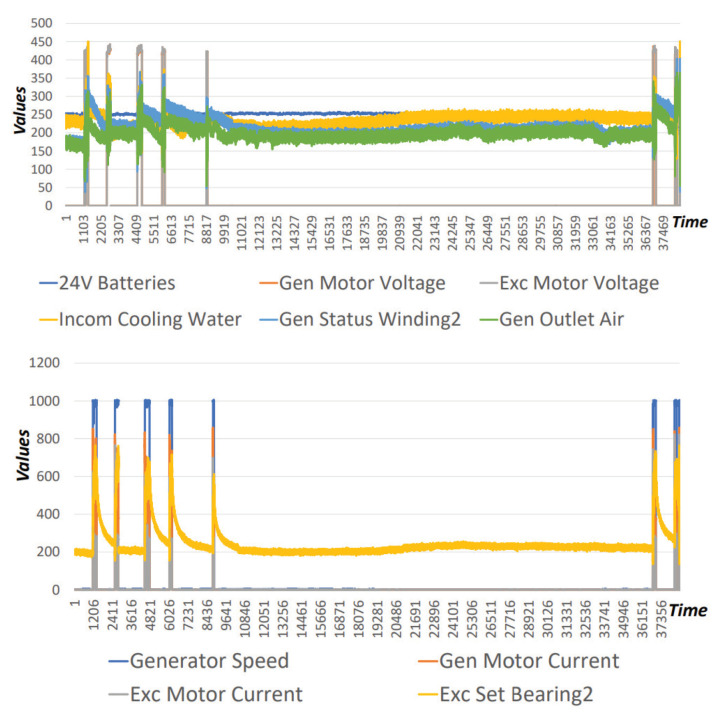
Operational Dataset (electricity measurements).

**Figure 6 sensors-20-05305-f006:**
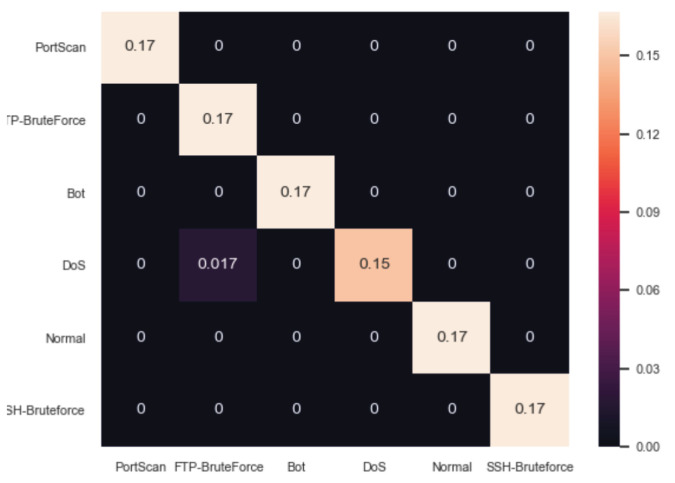
Confusion Matrix of IDM (Decision Tree Classifier).

**Figure 7 sensors-20-05305-f007:**
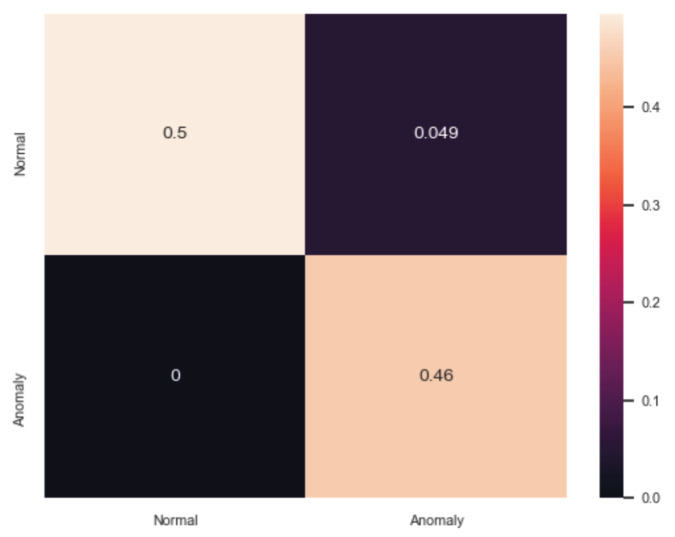
Confusion Matrix of ADM (Autoencoder).

**Figure 8 sensors-20-05305-f008:**
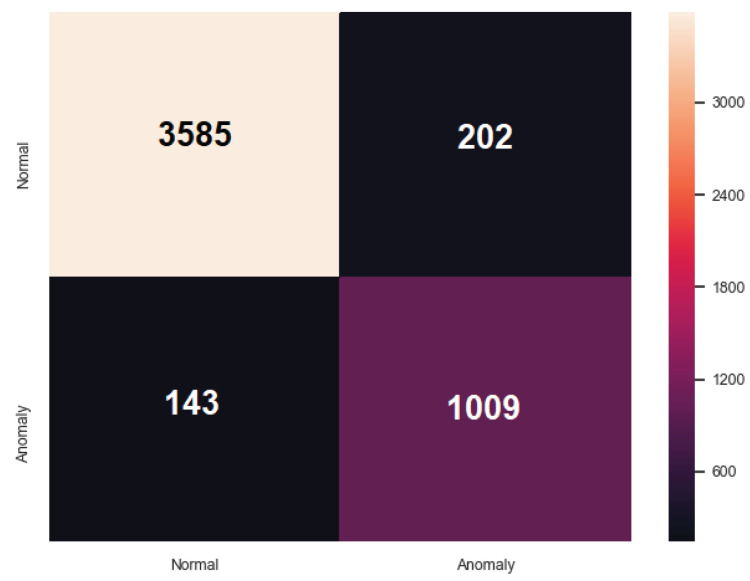
Confusion Matrix of ARIES Third Detection Layer (ARIES GAN).

**Table 1 sensors-20-05305-t001:** Evaluation Results of the First ARIES Detection Layer—IDM.

ML Method	Accuracy	TPR	FPR	F1
Logistic Regression	0.922	0.767	0.046	0.767
LDA	0.882	0.648	0.070	0.648
**Decision Tree Classifier**	**0.994**	**0.982**	**0.003**	**0.982**
Naive Bayes	0.917	0.751	0.049	0.751
SVM RBF	0.841	0.523	0.095	0.523
SVM Linear	0.802	0.406	0.118	0.406
Random Forest	0.990	0.970	0.005	0.970
MLP	0.909	0.728	0.054	0.728
AdaBoost	0.846	0.538	0.092	0.538
Quadratic Discriminant Analysis	0.722	0.166	0.166	0.166
Dense DNN Relu	0.984	0.954	0.009	0.954
Dense DNN Tanh	0.965	0.897	0.020	0.897

**Table 2 sensors-20-05305-t002:** Evaluation Results of the First ARIES Detection layer—ADM.

ML Method	Accuracy	TPR	FPR	F1
ABOD	0.944	1	0.101	0.942
Isolation Forest	0.938	0.999	0.111	0.937
PCA	0.545	0	0	0
LOF	0.944	1	0.101	0.942
MCD	0.493	0.001	0.097	0.002
**Autoencoder**	**0.9514**	**1**	**0.0963**	**0.9533**

**Table 3 sensors-20-05305-t003:** Evaluation Results of the Second ARIES Detection Layer

ML Method	Accuracy	TPR	FPR	F1
ABOD	0.581	0.993	0.522	0.487
**Isolation Forest**	**0.917**	**0.751**	**0.049**	**0.751**
PCA	0.745	0.978	0.312	0.606
MCD	0.733	0.210	0.135	0.240
LOF	0.733	0.210	0.135	0.240
Autoencoder	0.746	0.978	0.311	0.607

**Table 4 sensors-20-05305-t004:** Evaluation Results of the Third ARIES Detection Layer.

ML Method	Accuracy	TPR	FPR	F1
ABOD	0.692	0.989	0.397	0.600
Isolation Forest	0.813	0.960	0.231	0.705
PCA	0.851	0.982	0.187	0.755
MCD	0.715	0.299	0.158	0.329
LOF	0.829	0.992	0.220	0.730
Autoencoder	0.851	0.982	0.188	0.755
**ARIES GAN**	**0.930**	**0.875**	**0.053**	**0.853**

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
