# Peer review of "ARIES: A Novel Multivariate Intrusion Detection System for Smart Grid"

_sensors, 2020, doi:10.3390/s20185305_

Round 1

Reviewer 1 Report

The authors propose an IDS, called ARIES, by combining three detection layers: a) network flows b) Modbus packets and c) operational data based on GAN.

The proposed approach is interesting but there are some points that the authors should consider.

The authors should be better described the novelties of their approach with respect to existing ones.  Furthermore, the authors should provide more details and discussion about the obtained results.

I suggest to analyze also more recent approaches about the examined topics. In particular, I suggest to cite the following papers for security analysis:

1) Recognizing unexplained behavior in network traffic. In Network Science and Cybersecurity (pp. 39-62). Springer, New York, NY.

2) FORGE: a fake online repository generation engine for cyber deception. IEEE Transactions on Dependable and Secure Computing.

3)  Online Mining Intrusion Patterns from IDS Alerts. Applied Sciences10(8), 2983.  

Finally, I suggest to perform a linguistic revision.

Author Response

Response to Reviewers’ Comments

Title: Securing the Smart Grid: A Comprehensive Compilation of Intrusion Detection and Prevention Systems

Authors: Panagiotis Radoglou Grammatikis, Panagiotis Sarigiannidis, Georgios Efstathopoulos and Emmanouil Panaousis

Response to Reviewer #1:

Comment 1: The proposed approach is interesting but there are some points that the authors should consider. The authors should be better described the novelties of their approach with respect to existing ones.

Response 1: The authors would like to thank the reviewer for the comment. The authors proceeded to the following changes.

Changes/Additions regarding Comment 1: Section 2 “2. Related Work and Contribution” of the paper was revised entirely by including two subsections, namely “2.1 Related Work” and “2.2 Contributions”. The first subsection analyses previous similar works while the second subsection clearly outlines the contributions of this paper, taking into account the previous analysis of the first subsection. Moreover, a new figure (Fig. 1) and the corresponding description were added in order to depict more clearly the main contributions and novelties of this paper. The changes mentioned above are between the lines 85 and 233. In particular, the contributions of the paper are described between the lines 195 and 233.

Comment 2: Furthermore, the authors should provide more details and discussion about the obtained results.

Response 2: The authors would like to thank the reviewer for this comment. The authors revised section 5.5 entitled “Evaluation Results”, by providing more details regarding the detection performance of the proposed Intrusion Detection System called ARIES (smArt gRid Intrusion dEtection System). In particular, the following additions were carried out.

Changes/Additions regarding Comment 2: First, in section 5.3 named “5.3 Evaluation Results”, more details were given for each evaluation metric, thus understanding better the corresponding numerical values. For example, the Accuracy metric can be impartial when the training and testing datasets consist of an equal number of data samples for all classes. Moreover, for each detection layer of ARIES, we discussed in detail the Machine Learning (ML) or the Deep Learning (DL) methods that achieve the best and worst detection results, respectively. In particular, regarding the first detection layer of ARIES, and more specifically regarding the Intrusion Detection Model (IDM), the necessary implementation details of the Decision Tree Classifier and how it operates are provided since this classifier produces the best Accuracy, True Positive Rate (TPR), False Positive Rate (FPR) and the F1 score. The respective information was also included for the other layers of ARIES. Finally, the necessary references for each ML and DL methods were incorporated. The aforementioned additions are between the lines 403 and 471.

Comment 3: I suggest to analyze also more recent approaches about the examined topics. In particular, I suggest to cite the following papers for security analysis:

1) Recognizing unexplained behavior in network traffic. In Network Science and Cybersecurity (pp. 39-62). Springer, New York, NY.

2) FORGE: a fake online repository generation engine for cyber deception. IEEE Transactions on Dependable and Secure Computing.

3)  Online Mining Intrusion Patterns from IDS Alerts. Applied Sciences, 10(8), 2983.

Response 3: The authors would like to thank the reviewer for raising these valuable references. The authors included and discussed them in the paper.

Changes/Additions regarding Comment 3: More specifically, the aforementioned references were included and described in subsection 2.1 entitled “Related Work”. For each reference mentioned above, a dedicated paragraph has been included. These additions are between the lines 151 and 181.

Comment 4Finally, I suggest to perform a linguistic revision.

Response 4: The authors would like to thank the reviewer for this comment. The paper has been thoroughly revised in terms of writing errors.

Changes/Additions regarding Comment 4: The paper has been thoroughly revised in terms of writing errors. Typos, syntax and grammar errors were corrected.

Reviewer 2 Report

The paper presents an interesting and new method for intrusion detection based on anomaly identification. The methods is well described and the  comparisons with other methods reveal its benefits.

There are no evident errors in the presentation. 

The ARIES architecture is presented to a high level. More details, e.g. technologies that are (re-)used, will be helpful for the reader.  

Author Response

Response to Reviewers’ Comments

Title: Securing the Smart Grid: A Comprehensive Compilation of Intrusion Detection and Prevention Systems

Authors: Panagiotis Radoglou Grammatikis, Panagiotis Sarigiannidis, Georgios Efstathopoulos and Emmanouil Panaousis

Response to Reviewer: #2

Comment 1: The ARIES architecture is presented to a high level. More details, e.g. technologies that are (re-)used, will be helpful for the reader.

Response 1: The authors would like to thank the reviewer for this comment. The following changes were made.

Changes/Additions regarding Comment 1: A new figure (Figure 3 - ARIES Technologies) along with the corresponding description were added by illustrating more specifically the technologies used by ARIES. These additions are between the lines 242 and 244. Moreover, the description of each architectural component of the proposed IDS includes a brief description of the respective technologies and how they are used by the corresponding components.

Reviewer 3 Report

  1. The authors explain that this article combines three detection layers: a) network flow-based detection, b) packet-based detection and c) operational data-based detection. From the perspective of the article, the three layers can operate and detect independently , There is no dependency. And from the description of the article, the contribution of this article is mainly in the third part. Please explain in detail the connection between the three detection layers and the main contribution of this article.
  2. The authors have provided some experimental data in Table 2, but the calculations of TP, TN, FP, FN in Figure 5 and the equations (3), (5), (6) in this article seem to be inconsistent with the values of Accuracy and F1 score in the Autoencoder in Table 2, please confirm or provide an example.
  3. The authors explain that this article has designed an IDS system. If an abnormality is found in the third detection layer in the system, what is the processing method? It can only remind users of system abnormalities or show the types of abnormalities that can be predicted.
  4. There are some typos in the article, please check carefully. For example, in line 418, "ARIES ...... Syste,".

Author Response

Response to Reviewers’ Comments

Title: Securing the Smart Grid: A Comprehensive Compilation of Intrusion Detection and Prevention Systems

Authors: Panagiotis Radoglou Grammatikis, Panagiotis Sarigiannidis, Georgios Efstathopoulos and Emmanouil Panaousis

Response to Reviewer: #3

Comment 1: The authors explain that this article combines three detection layers: a) network flow-based detection, b) packet-based detection and c) operational data-based detection. From the perspective of the article, the three layers can operate and detect independently, There is no dependency. And from the description of the article, the contribution of this article is mainly in the third part. Please explain in detail the connection between the three detection layers and the main contribution of this article.

Response 1: The authors would like to thank the reviewer for this comment. The following changes were made.

Changes/Additions regarding Comment 1: First, regarding the contributions of the paper, Section 2 entitled “Related work and Contributions” was revised by including now two subsections, namely “2.1 Related Work” and “2.2 Contributions”. As indicated by its name, the second subsection (“2.2 Contributions”) discusses and enumerates the paper's contributions and novelties. In addition, as denoted in line 222, the main contribution of the paper is the proposed Generative Adversarial Network (GAN), which is applied at the third layer of ARIES. These changes are located between the lines 196 and 233. Finally, the same subsection (line 197) clarifies that the operation of the three detection layers is independent.

Comment 2: The authors have provided some experimental data in Table 2, but the calculations of TP, TN, FP, FN in Figure 5 and the equations (3), (5), (6) in this article seem to be inconsistent with the values of Accuracy and F1 score in the Autoencoder in Table 2, please confirm or provide an example.

Response 2: The authors thank the reviewer for this significant note. The necessary changes were carried out.

Changes/Additions regarding Comment 2: The numerical results of all tables and figures were checked again. The specific entry of Table 2 (“Evaluation Results of the First ARIES Detection layer - ADM”) was corrected. Now, the autoencoder is characterized by the following metrics. It is noteworthy, that the current results are better than the previous ones.

0.0963

Comment 3: The authors explain that this article has designed an IDS system. If an abnormality is found in the third detection layer in the system, what is the processing method? It can only remind users of system abnormalities or show the types of abnormalities that can be predicted.

Response 3: The authors thank the reviewer for this significant note. The necessary changes were carried out.

Changes/Additions regarding Comment 3: The functionality and the corresponding description of the ARIES Response Module (subsection 3.3 – “3.3 Response Module”) were improved so that iptables can apply firewall rules based also on the security events related to the third layer. The firewall rules are constructed in terms of IP addresses and the Transmission Control Protocol (TCP)/User Datagram Protocol (UDP) ports. The specific change is between the lines 288 and 294.

Comment 4: There are some typos in the article, please check carefully. For example, in line 418, "ARIES ...... Syste,".

Response 4: The authors would like to thank the reviewer for this comment. The paper has been thoroughly revised in terms of writing errors.

Changes/Additions regarding Comment 4: The paper has been thoroughly revised in terms of writing errors. Typos, syntax and grammar errors were corrected.

Round 2

Reviewer 1 Report

I think that the authors have addressed all my concerns. 

Reviewer 3 Report

The authors have corrected the previous questions, but in Lines 231-233 the authors state that this article evaluates a large number of DL/ML methods, but in tables 1,2,3,4 the reader does not find any solutions using deep learning DL. The authors should have added a comparison with deep learning related methods, not just with machine learning subject methods.
